# Anterior Umbilication of Lens in a Family with Congenital Cataracts Associated with a Missense Mutation of *MIP* Gene

**DOI:** 10.3390/genes13111987

**Published:** 2022-10-31

**Authors:** Zhixing Cheng, Xun Wang, Qiwei Wang, Xulin Zhang, Dongni Wang, Weiming Huang, Meimei Dongye, Xiaocheng Feng, Danying Zheng, Haotian Lin

**Affiliations:** 1Guangdong Eye Institute, Department of Ophthalmology, Guangdong Provincial People’s Hospital, Guangdong Academy of Medical Sciences, Guangzhou 510080, China; 2State Key Laboratory of Ophthalmology, Zhongshan Ophthalmic Center, Sun Yat-sen University, Guangzhou 510060, China; 3People’s Hospital of Yingde City, Qingyuan 513000, China

**Keywords:** major intrinsic protein (*MIP*), congenital cataracts (CCs), anterior umbilication of the lens, whole-exome sequencing (WES), elongated axial length

## Abstract

Congenital cataracts (CCs) have significant genotypic and phenotypic heterogeneity. The major intrinsic protein (*MIP*) gene, one of the causative genes of CCs, plays a vital role in maintaining the homeostasis and transparency of the lens. In this study, we identified a unique phenotype of anterior umbilication of the lens in a four-generation pedigree with CCs. All patients in the observed family had nystagmus, nuclear cataracts, and elongated axial lengths compared with their healthy counterparts except for patient I:2, whose axial length was unavailable, and patientII:4, who had total cataracts. We confirmed, using Sanger sequencing based on whole-exon sequencing (WES) data, that all patients carried a heterozygous variant NM_012064.4:c.97C > T (NP_036196.1:p.R33C) in their *MIP* gene. To our knowledge, 29 variants of the human *MIP* gene and the relative phenotypes associated with CCs have been identified. Nevertheless, this is the first report on the anterior umbilication of the lens with nuclear or total opacity caused by the c.97C > T (p.R33C) variant in the *MIP* gene. These results also provide evidence that the elongated axial length might be associated with this variant. This study further confirms the phenotypic heterogeneity of CCs.

## 1. Introduction

Congenital cataracts (CCs) are an avoidable cause of visual impairment or blindness, with the overall prevalence ranging from 0.006% to 0.097% worldwide [1]. Approximately 30% of CCs are caused by single-gene variants [2]. The inheritance patterns mainly include autosomal dominant inheritance, autosomal recessive inheritance, and X-linked inheritance [3]. Significant genotypic and phenotypic heterogeneity is observed in CCs. The clinical phenotypes of CCs are complex and diverse and include anterior subcapsular, posterior polar, posterior subcapsular, nuclear, cortical, lamellar, zonular, pulverulent, cerulean, sutural, polymorphic, membranous, and total cataracts [4]. The major intrinsic protein (*MIP*) gene is a critical membrane protein gene in the ocular lens that encodes the *MIP*, also known as aquaporin0 (*AQP0*), which plays a vital role in maintaining the integrity and transparency of the lens [5]. The *MIP* is embedded in the plasma membrane with six transmembrane bilayer-spanning domains (H1H6), three extracellular loops (A, C, and E), two intracellular loops (B and D), and NH2− and COOH− terminal intracellular domains (N-/C-TIDs). At present, 29 variants in the human *MIP* gene associated with CCs have been identified, and most of the variants are located in the H4, H5, H6, C-TIDs, and loop C of the *MIP* [6]. The NP_036196.1:p.R33C variant located in the extracellular loop A and the clinical phenotypes caused by this variant previously reported in different families with CCs included total cataract and nuclear cataract with posterior polar opacity [7,8,9,10,11]. Herein, we found a novel cataract phenotype, anterior umbilication of the lens, and nuclear or total cataract in a Han Chinese family with CCs caused by the p.R33C variant in the *MIP* gene.

## 2. Materials and Methods

### 2.1. Patients

Eight affected members from a four-generation Chinese family were recruited in this study. The present study was conducted according to the Declaration of Helsinki and approved by the Institutional Review Board of Zhongshan Ophthalmic Center (no. 2020KYPJ007). All patients or their guardians signed written informed consent forms.

### 2.2. Clinical Assessments

We evaluated all available participants’ clinical and family histories in this four-generation family. We performed comprehensive ophthalmic examinations, including the best-corrected visual acuity (BCVA) assessment, intraocular pressure (IOP), slit-lamp biomicroscope, anterior segment (AS) and fundus photography (TRC-NW400, Topcon Inc., Tokyo, Japan), B-ultrasound scan (Aviso, Quantel Medical, Clermont-Ferrand, France), A-ultrasound (Aviso, Quantel Medical, Clermont-Ferrand, France), optical coherence tomography (OCT) and anterior segment OCT (AS-OCT) (Optovue, RTVue XR Avanti, Optovue, Inc., Fremont, CA, USA), and IOL Master (IOL Master 700, Zeiss, Carl Zeiss Meditec AG, Jena, Germany). A-ultrasound scan or IOL Master was applied to measure the axial length. Detailed lens phenotypes were recorded by anterior segment photography and AS-OCT.

### 2.3. Genetic Variant Screening

Genomic DNA was extracted from peripheral blood samples collected from the participants as previously reported [12]. The genetic variant was detected with the whole-exome sequencing (WES) technique using an Agilent v6 exome (Agilent Technologies, Inc., Santa Clara, CA, USA) and the Illumina platform (Illumina Inc., CA, USA) following a quality inspection. Alignment against the human reference genome (NCBI build 37/hg19) was performed to map sequence reads using CLC Genomics Workbench (version 6.5.2) software (CLC bio, Aarhus, Denmark). The gnomAD databases (https://gnomad.broadinstitute.org, accessed on 30 June 2022, 141,456 individuals; January 2022 data release), and the 1000 Genomes database (http://grch37.ensembl.org/Homo_sapiens/Info/Index, accessed on 30 June 2022, September 2015 data release) were used to determine the frequencies of variants, and variants with minor allele frequency (MAF) values > 0.0002 were removed. Prediction tools, including PolyPhen-2 (http://genetics.bwh.harvard.edu/pph2, accessed on 30 June 2022) [13], SIFT (http://sift.jcvi.org, accessed on 30 June 2022) [14], and MutationTaster (version number: MutationTaster2, https://www.mutationtaster.org, accessed on 30 June 2022) [15] online tools, were utilized to predict the pathogenicity of missense variants. The splice site prediction program with a neural network (BDGP, https://www.fruitfly.org/seq_tools/splice.html, accessed on 30 June 2022) was used to predict the pathogenicity of splicing variants in the synonymous and introns variants [16]. Candidate variants were further confirmed by Sanger sequencing. Segregation analyses were performed for all available members of the family. Moreover, sequence variants were classified based on the American College of Medical Genetics and Genomics (ACMG) guidelines [17].

## 3. Results

### 3.1. Clinical Findings

Eight patients (seven female and one male) from a four-generation Chinese family were investigated in this study (Figure 1A). All patients were diagnosed with bilateral CCs and underwent cataract surgery with or without (IV:6) intraocular lens implant. The clinical manifestations of the patients included severe visual impairment, nuclear opacity, total opacity, the anterior umbilication of the lens, elongated axial length, and nystagmus. The cataract morphology of all patients was nuclear opacity except for II:4, whose cataract morphology was total opacity. Three (II:4, III:4, and III:6) patients had obvious anterior umbilication of the lens. AS-OCT showed an obvious umbilication in the central zone of the lens with an intact anterior capsular membrane. Moreover, III:4 had spontaneous subluxation of the lens and sensory esotropia in his left eye (Figure 2E,F). III:6 also had spontaneous subluxation of the lens. The axial length of II:4 (32.35 mm) was clearly longer than those of III:4 (26.85 mm) and III:6 (26.84 mm). In patients III:4 and III:6, it is noteworthy that the axial lengths were rapidly elongated with age, and the axial lengths of the left eyes were longer than those of the right eyes (Table 1). There was no family history of other ocular or systemic disorders.

### 3.2. Molecular Findings and Bioinformatics Analysis

A heterozygous missense variant NM_012064.4:c.97C > T (NP_036196.1:p.R33C) in exon 1 of the *MIP* gene was identified in all affected individuals tested (Figure 1B). It was not detected in unaffected family members, the 1000 Genomes database, or the gnomAD databases. The variant causes a substitution of arginine (R) for cysteine (C) at codon 33 in extracellular loop A of the *MIP* protein. Bioinformatics analysis with PolyPhen-2, SIFT, and MutationTaster revealed that the substitution of arginine for cysteine at codon 33 in the *MIP* gene was predicted as possibly being damaging and disease-causing. ACMG classification of this variant was likely pathogenic considering that it met two moderate pieces of evidence (PM1 and PM2) and two supporting pieces of evidence (PP1 and PP3) [17]. The detailed classification basis is as follows: (1) PM1: this variant is located in the first loop outside the transmembrane portion of *MIP*. The pathogenicity of the variant for congenital cataracts has been verified in the transgenic embryonic chick model [18]. (2) PM2: this variant is absent in the gnomAD and the 1000 Genomes databases. (3) PP1: this variant is co-segregated with CCs in all available members of this family. Moreover, there were more segregation data for this variant, including families from China, Australia, Cambodia, and Switzerland [7,8,9,10,11], which provided stronger pathogenic evidence. (4) PP3: all the in-silico programs (PolyPhen-2, SIFT, and MutationTaster) support a deleterious effect of this variant.

## 4. Discussion

We found a novel phenotype of CCs associated with a missense variant NM_012064.4:c.97C > T (NP_036196.1:p.R33C) in the *MIP* gene. As the causative variant of CCs, the p.R33C variant has previously been reported in several families with CCs [7,8,9,10,11]. This variant was first reported by Gu et.al in 2007 and the phenotype was bilateral total cataract manifested as complete opacification of the fetal nucleus and the cortex [7]. However, the phenotype reported by Zhou et.al was a bilateral nuclear cataract characterized as a central nuclear opacity involving the embryonic and fetal nuclei with posterior polar opacity [8]. In this study, the phenotype was typical anterior umbilication of the lens in three affected members with nuclear or total cataracts. To our knowledge, this was the first case to report anterior umbilication of the lens in patients with congenital cataracts. The umbilication was located at the anterior surface and central zone of the lens. The phenotypes of CCs in this family included anterior umbilication of the lens, nuclear cataract, and total cataract. The results provided strong evidence to support the phenotypic heterogeneity of variants in *MIP* in patients with CCs. Moreover, patients III:4 and III:6 had significant subluxation of the lens in their left eyes. However, it is unclear whether the subluxation of the lens or the abnormalities of the zonular fibers was associated with this variant.

All the affected members had elongated axial lengths compared with their healthy counterparts in the same age range except for patient I:2, as her axial length was unavailable. These results suggest that the elongated axial length was associated with this variant. This is consistent with the results of a previous study [7]. It is noteworthy that the axial length of patient II:4 was much longer than those of her children (III:4 and III:6). Additionally, the latest axial length of II:4 was longer than that measured eight years ago. These results suggest that the axial length was probably elongated with age. Moreover, the axial lengths of the children’s (III:4 and III:6) left eyes were both longer than those of their right eyes. Due to their left eyes having nuclear cataracts, continuous form deprivation may contribute to the elongated axial length. In other words, the timely removal of cataracts may contribute to controlling axial length growth. Moreover, all affected individuals in this family showed nystagmus caused by long-term visual deprivation.

The *MIP*, composed of approximately 50% of the total fiber cell membrane proteins in the lens, is involved in lens metabolism and osmotic pressure regulation by transporting water to the cell membrane [19]. Furthermore, *MIPs* act as adhesion molecules to maintain intact junctional structures between lens fiber cells [20]. Kumari et al. demonstrated that the p.R33C variant of *AQP0* caused a significant reduction in the cell-to-cell adhesion of fiber cells but did not influence the water channel function [21]. Lo et al. found that the interlocking protrusions of the cortical fiber cells in a *MIP*-deficient lens had undergone deformation and fragmentation, which resulted in fiber cell separation and breakdown and nuclear cataract formation [5]. These results suggest that *MIP*/*AQP0* is vital for lens transparency and homeostasis.

It is certain that any changes in the *MIP* may lead to cataract formation in humans and mice. However, the formation mechanism of the anterior umbilication of the lens is unclear. According to the AS-OCT results, although the anterior surface of the lens had a deep invagination with a steep slope, the anterior capsular membrane of the lens was intact. We presume that the possible reasons for the umbilication formation are as follows: (i) the degradation and absorption of the abnormal cortical fiber cells in the anterior polar region led to the local umbilication; (ii) the lens development process, in which the lens epithelial cells in the equatorial region continuously divide and differentiate into new fiber cells during postnatal lens development, was abnormally ceased at a specific timepoint; and (iii) the highly ordered arrangement of the central cortical fiber cells were destroyed due to the loss of the interlocking protrusions. All four affected members of the fourth generation had dense nuclear opacity without anterior umbilication. The size of the anterior umbilication in patients III:4 and III:6 was almost the same as the size of their pupils in the daytime. However, the size of the anterior umbilication in patient II:4 was much bigger than that of her children (III:4 and III:6). These results suggest that the anterior umbilication of the lens is possibly progressive. Interestingly, the shape and size of the anterior umbilication in the right eye (photographed in 2013) of patient II:4 was nearly the same as her left eye (photographed in 2021). This finding suggests that the anterior umbilication of the lens was possibly static or stopped progressing at a certain age. These results seem to be paradoxical. In fact, these results further indicate the phenotypic heterogeneity of congenital cataracts. It is noteworthy that the posterior umbilication of the lens only occurred in the formalin-fixed lens and was not present in vivo [22]. Therefore, the posterior umbilication of the lens was not considered a lens disorder and had no clinical significance. In this study, we first described a phenotype of anterior umbilication of the lens as a lens disorder of congenital cataracts in detail. It might shed light on the pathogenesis of cataracts in patients with the c.97C > T (p.R33C) variant in the *MIP*.

## 5. Conclusions

The anterior umbilication of the lens is a novel cataract phenotype that has not been reported previously. This study provided the first genetic evidence exploring the pathogenesis of the anterior umbilication of the lens. Further studies are necessary to confirm the associations and define the exact mechanism.

## Figures and Tables

**Figure 1 genes-13-01987-f001:**
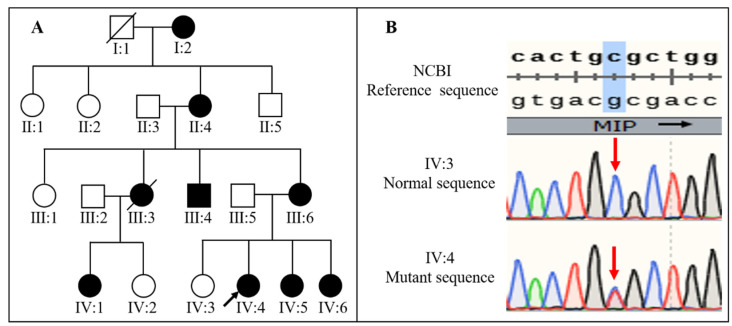
The pedigree of a four-generation family and sequencing results of the *MIP* genetic variant. (**A**) In the pedigree of a congenital cataract family with variant NP_036196.1:p.R33C of the *MIP* gene, the black arrow indicates the proband. (**B**) Sequence maps of the involved sequence fragment of the *MIP* show a heterozygous C > T transition at position 97 (red arrow). IV:4, the proband, IV:3, the unaffected member.

**Figure 2 genes-13-01987-f002:**
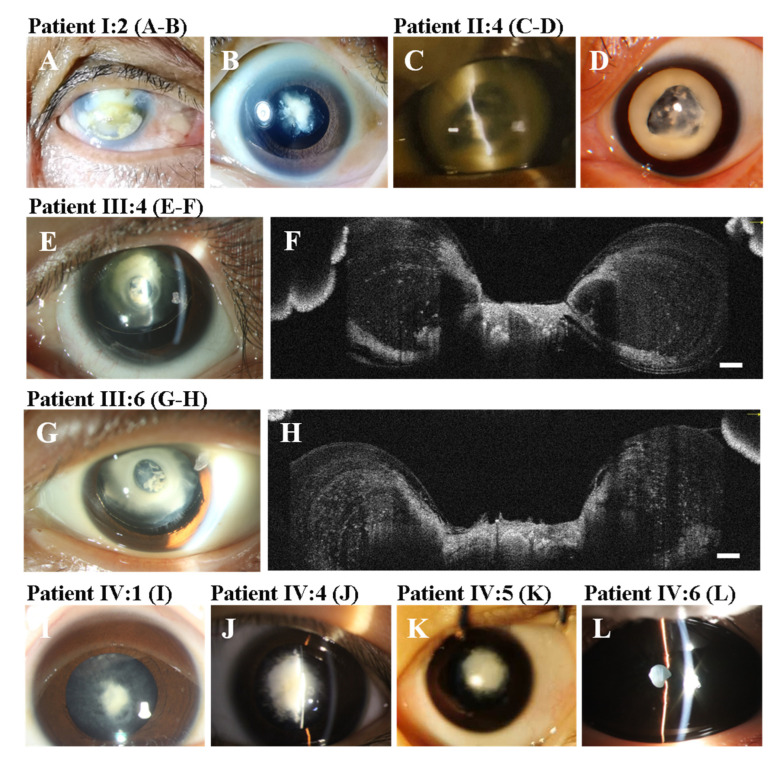
The clinical findings of the affected members (**A**) I:2, keratoleukoma, od; (**B**) I:2, nuclear cataract, os; (**C**,**D**) II:4, total cataract and anterior umbilication of the lens, C, od (2013), D, os (2021); (**E**,**F**) III:4, the anterior umbilication of the lens, nuclear cataract, and subluxation of the lens. AS-OCT showed the anterior umbilication of the lens; (**G**,**H**) III:6 the anterior umbilication of the lens, nuclear cataract, and subluxation of the lens. AS-OCT showed the anterior umbilication of the lens; (**I**) IV:1 nuclear cataract; (**J**) IV:4, the proband, nuclear cataract; (**K**), IV:5, nuclear cataract; and (**L**) IV:6, nuclear cataract. Scale bar: 250 μm.

**Table 1 genes-13-01987-t001:** Summary of clinical features and genetic variants of patients with congenital cataracts.

Patients	I:2	II:4	III:4	III:6	IV:1	IV:4	IV:5	IV:6
Sex	F	F	M	F	F	F	F	F
Age	71 ys	49 ys	27 ys	25 ys	5 ys 8 ms	5 ys	3 ys 2 ms	1 y 2 ms
BCVA * (logMAR **, od/os)	NLP/FC	1.00/FC	1.00/HM	1.00/HM	HM/HM	HM/HM	HM/HM	NA
Variant site	c.97C > T	c.97C > T	c.97C > T	c.97C > T	c.97C > T	c.97C > T	c.97C > T	c.97C > T
Amino acid change	p.R33C	p.R33C	p.R33C	p.R33C	p.R33C	p.R33C	p.R33C	p.R33C
Cataract morphology	Nuclear opacity	Total opacity	Nuclear opacity	Nuclear opacity	Nuclear opacity	Nuclear opacity	Nuclear opacity	Nuclear opacity
Anterior umbilication of lens	−	+	+	+	−	−	−	−
Subluxation of lens	−	−	+	+	−	−	−	−
Cataract surgery	od	od	od	od	ou	ou	ou	ou
Date of surgery	1966-06	2013-06	2013-06	2013-06	2020-10	2020-12	2021-01	2021-01
Pre-op axial length (mm, od/os)	NA	32.35/NA	26.85/NA	26.84/NA	25.33/25.03	23.54/23.52	21.75/21.34	17.22/17.17
Latest axial length (mm, od/os)	NA	32.74/NA	31.52/31.96	31.08/31.20	NA	NA	NA	NA
Nystagmus	+	+	+	+	+	+	+	+
Complications	Leukoma od	PCO od	PCO, RRD od	PCO, RRD od	NA	NA	NA	NA

Abbreviations: F, female; M, male; ys, years; y, year; ms, months; BCVA, best-correct visual acuity; logMAR, the logarithm of the minimum angle of resolution; od, the right eye; os, the left eye; ou, both eyes; NLP, no light perception; FC, fingers counting; HM, hand motion; Pre-op, pre-operation; NA, not available; +, present; −, absent; PCO, posterior capsular opacity; RRD, rhegmatogenous retinal detachment. * BCVA, The best-correct visual acuity of patients with very low vision (logMAR > 1.00) was classified using the semi-quantitative clinical scale “fingers counting” (FC), “hand motion” (HM), “light perception” (LP), and “no light perception” (NLP). ** logMAR, ranging from −0.30 to 1.00, smaller values mean better visual acuity in general.

## Data Availability

The data are available from the corresponding authors upon reasonable request.

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
