# Peer review of "Anterior Umbilication of Lens in a Family with Congenital Cataracts Associated with a Missense Mutation of MIP Gene"

_genes, 2022, doi:10.3390/genes13111987_

Round 1

Reviewer 1 Report

The authors presented a very interesting study reporting for the first time anterior umbilication of the lens in a congenital cataract family caused by the c.97C>T (p.R33C) mutation in the MIP gene. The manuscript is clear, its topic is original in content, and the conclusions are consistent with the evidence presented.  The manuscript is with merit and the findings are worth reporting - my only comment is that the authors should improve the quality of presentation of the manuscript and revise it for presence of typos, errors in punctuation, use of abbreviations in the legends of the figures without corresponding explanations, and in general English style.

Author Response

Point 1: My only comment is that the authors should improve the quality of presentation of the manuscript and revise it for presence of typos, errors in punctuation, use of abbreviations in the legends of the figures without corresponding explanations, and in general English style.

Response 1: Thanks a lot for your advice. As suggested, for the convenience of readers' understanding of ophthalmic terminology, we have added two annotations to abbreviations in table 1. The text has undergone English language editing by MDPI now and here is the certification of the manuscript.

Reviewer 2 Report

In this manuscript by Cheng et al., the authors describe phenotype associated with congenital cataract in a family affected by p.R33C mutation in the MIP gene. The authors report anterior umbilication of the lens in at least two of the affected patients and suggest that this mutation is associated with phenotypic heterogeneity.

Minor comments on the manuscript:

1.       Figure 2: It would be helpful to add Patient numbers to each panel. For example: Include I:2 label in figure 2A, 2; II:4 in figure 2C, 2D etc. This would make it much easier for readers to correlate images with patient numbers.

2.       Table 1: The authors have explained abbreviations FC: Finger count and HM: Hand motion, however, it would be useful to explain what they mean. Does HM mean that the patients cannot detect hand motion?

3. Line 156: It would be good to use consistent nomenclature for amino acid changes. So using p.R130C instead of p.Arg130Cys to match with the format used elsewhere in the manuscript would be best. 

Author Response

Point 1: Figure 2: It would be helpful to add Patient numbers to each panel. For example: Include I:2 label in figure 2A, 2; II:4 in figure 2C, 2D etc. This would make it much easier for readers to correlate images with patient numbers

Response 1: Thanks a lot for the suggestion. We have fixed the patients label issue for each figure.

Point 2: Table 1: The authors have explained abbreviations FC: Finger count and HM: Hand motion, however, it would be useful to explain what they mean. Does HM mean that the patients cannot detect hand motion?

Response 2: We thank the reviewer for pointing this out. For the convenience of readers' understanding of ophthalmic terminology, we have added two annotations to table 1. In clinical, the visual acuity (VA) of patients with very low vision is classified using the semi-quantitative clinical scale " fingers counting" (FC), "hand motion" (HM), "light perception" (LP) and "no light perception" (NLP). “Fingers counting”(FC) means patients can not detect any figures in the visual chart but still the finger number in the distance of 5 cm to 1m. “Hand motion”(HM) means patients can even detect no fingers but only the hand within 5cm.

Point 3: Line 156: It would be good to use consistent nomenclature for amino acid changes. So using p.R130C instead of p.Arg130Cys to match with the format used elsewhere in the manuscript would be best.

Response 3: As suggested, all the nomenclature for amino acid changes in the manuscript has been adjusted to the same expression form. (Line 160, highlighted in yellow)